# AI for Interpretable Chemistry: Predicting Radical Mechanistic Pathways via Contrastive Learning

**Mohammadamin Tavakoli**
Department of Computer Science
University of California, Irvine
mohamadt@uci.edu

**Yin Ting T.Chiu**
Department of Chemistry
University of California, Irvine
yintc@uci.edu

**Alexander Shmakov**
Department of Computer Science
University of California, Irvine
ashmakov@uci.edu

**Ann Marie Carlton**
Department of Chemistry
University of California, Irvine
agcarlto@uci.edu

**David Van Vranken**
Department of Chemistry
University of California, Irvine
david.vv@uci.edu

**Pierre Baldi**
Department of Computer Science
University of California, Irvine
pfbaldi@uci.edu

## Abstract

Deep learning-based reaction predictors have undergone significant architectural evolution. However, their reliance on reactions from the US Patent Office results in a lack of interpretable predictions and limited generalizability to other chemistry domains, such as radical and atmospheric chemistry. To address these challenges, we introduce a new reaction predictor system, RMechRP, that leverages contrastive learning in conjunction with mechanistic pathways, the most interpretable representation of chemical reactions. Specifically designed for radical reactions, RMechRP provides different levels of interpretation of chemical reactions. We develop and train multiple deep-learning models using RMechDB, a public database of radical reactions, to establish the first benchmark for predicting radical reactions. Our results demonstrate the effectiveness of RMechRP in providing accurate and interpretable predictions of radical reactions, and its potential for various applications in atmospheric chemistry.

## 1 Introduction

Three primary approaches have been employed for the systematic prediction of chemical reactions: Quantum Mechanics simulations [1; 2], Template-based methods [3; 4; 5; 6; 7; 8], and machine learning methods [9; 10; 11; 12; 13; 14; 15]. Notably, among these methods, machine learning techniques have demonstrated superior generalization capabilities and have enabled high-throughput predictions.

However, the majority of recently developed data-driven predictive models are primarily designed to operate at the level of overall transformations. These predictors are trained using the chemical transformation dataset sourced from the US Patent Office (USPTO) [16], along with a few additional, though relatively small, datasets introduced scatteredly in various studies [17; 18; 19]. It is important to note that all these datasets have notable limitations, which include: limited coverage of chemical reactions, lack of information about the byproducts, an imbalance between reactants and products,

and a scarcity of information concerning elementary reaction steps. For example, the USPTO dataset predominantly presents chemical reactions as overall transformations with a single primary product, providing limited insights into underlying mechanisms, critical intermediates, and side products. Moreover, extracting information on radical reactions is challenging, as they are underrepresented in this dataset.

The utilization of this constrained source of data for training data-driven models leads to predictive models that may have inherent limitations and biases, primarily catering to specific types of chemical reactions. These models often struggle to predict balanced reactions and lack crucial information regarding intermediate byproducts. Moreover, they fall short in offering explanations regarding the underlying chemistry responsible for the predicted products. Given the scarcity of available training data, there is currently no widely adopted reaction predictor specifically tailored for radical reactions. Radical reactions play a significant role in various domains, including synthetic pathway planning, biological chemistry, and atmospheric chemistry. These reactions frequently involve intricate sequences of chemical steps and highly branched mechanistic pathways. Hence, there is a critical need to develop an accurate radical reaction predictor that can effectively address the aforementioned limitations.

## 2    Interpretability of Mechanistic Reaction Steps

Instead of overall transformations (e.g., samples of the USPTO dataset), expert chemists typically employ an alternative approach to represent and conceptualize chemical reactions. Specifically, they utilize an intuitive representation that involves the consideration of single-state transitions and arrow-pushing mechanisms, which we refer to as elementary reaction mechanisms or elementary steps. Overall chemical transformations can be deconstructed into a chain of elementary reaction mechanisms, each characterized by a singular transition state [20; 21]. Figure 1 provides an illustrative example of this process. Curved arrows (or fish-hook arrows for radical reactions) are employed to depict the reaction mechanisms [22], and correspond to the interaction of singly occupied molecular orbitals with both the highest occupied molecular orbital (HOMO) and the lowest unoccupied molecular orbital (LUMO) [23]. Hence, the development of a reaction predictor that operates at the reaction mechanism level can confer three critical benefits (shown in Figure 1 that none of the current reaction predictors can offer.

**Chemical interpretability:** The first key benefit is enhanced chemical/orbital interpretability. The use of curly arrows, or arrow-pushing mechanisms, allows for an accurate understanding of the fundamental chemistry underlying each reaction step. This approach facilitates the understanding of the interactions between molecular orbitals, which ultimately drive each reaction step.

**Pathway interpretability:** The second key benefit is enhanced pathway/transformation interpretability. A predictor trained to predict elementary steps can be iterated to expand a tree of such steps rooted at the initial reactants. This allows for the interpretation of any overall transformations leading to several final products, some of which are unknown. By expanding such a tree of pathways there is no chance of missing key intermediates that give rise to competing pathways during change. This level of interpretability enables several applications, most importantly for drug discovery and atmospheric chemistry where highly reactive radicals are involved in massively branched reactions.

**Balance and atom mapping:** Finally, the third benefit is the preservation of the balance between reactants and products, in conjunction with the underlying atom mapping. The balance is maintained at all times throughout the tree of pathways (i.e., the chain of reaction steps), which can be highly valuable, for instance in retrosynthesis [24], and mass spectrometry.

## 3    Data

To develop a radical reaction predictor operating at the mechanistic steps, we utilize the standard train and test sets from the recently released RMechDB dataset [25]. This dataset comprises approximately 5500 radical elementary step reactions sourced from chemistry textbooks (core reactions) and scientific articles on atmospheric chemistry (atmospheric reactions). Each reaction in the RMechDB dataset is labeled with different categorizations enabling comprehensive evaluation of the predictive models trained on RMechDB. See Table 1 for a summary of the RMechDB dataset used in subsequent training and testing experiments.

Figure 1: The reaction at the top is an overall, unbalanced, transformation from the USPTO dataset. It can be broken down into four mechanistic steps with arrow-pushing mechanisms. This provides chemical interpretability for each step, as well as for the overall pathway while maintaining full balance at each step.

Table 1: The size of the various subsets of elementary radical reaction steps contained in the RMechDBdatabase [25]. These are used to train and test the predictors.

|  | Train | Test |
|---|---|---|
| Core | 1512 | 150 |
| Specific | 3397 | 367 |
| Combined | 4909 | 517 |

## 4 Methods

Here we first describe the *OrbChain* as a model of radical mechanistic reaction based on the interaction of idealized molecular orbitals. We utilize Orbchain to represent and process radical mechanistic reactions for machine learning models. Then we describe three different machine learning approaches for predicting the outcome of radical mechanistic reactions with or without their associated arrow-pushing mechanisms. The first approach follows the methodology described in [20; 21; 26] where predictions are carried out using deep learning in two steps. The first step identifies reactive sites, while the second step ranks the plausibility of all possible reactions based on the interactions of the previously identified reactive sites. In the second approach, we directly predict the most reactive pair of molecular orbital. Within this one-step prediction approach, we test two different representations for the atoms: our own atom descriptor and RxnHypergraph [27] which applies a transformer model to the molecular graph representations. Lastly, we use a purely text-based approach [28] leveraging transformers and large language models to predict the products of radical reactions using a neural machine translation system [29]. This approach is based on the text representation of chemical reaction and aims to "translate" the sequence of reactants into the sequence of products. Notably, this approach only outputs the string associated with the products, without the arrow-pushing information. As a result, this approach cannot provide chemical interpretability, although it shows promising results on the RMechDB test sets.

### 4.1 OrbChain: Standard Model of Mechanistic Reactions

A radical mechanistic reaction Rxn is a reaction with a single transition state that involves at least one half-occupied orbital. Rxn consists of a set of reactant molecules $R = \{R_i\}_{i=1}^{n_r}$, a set of product molecules $P = \{P_i\}_{i=1}^{n_p}$, and a set of fish-hook arrows $A$ showing a bond cleavage and movements of a single electrons. Each of the reactant and product molecules $R_i$, $P_i$ is represented by a connected molecular graph $G_i = (N_i, V_i)$ where the vertices $N_i$ represent labeled atoms $\{a_j^i\}$ and edges $V_i$ represent labeled bonds $\{b_j^i\}$. Inspired by [20; 21], we model a radical mechanism by OrbChain as

the interaction between two reactive Molecular Orbitals (MOs) $m_1^{(*)}$ and $m_2^{(*)}$ (we refer to an MO as $m$ and a reactive MO as $m^{(*)}$). This reaction model matches the arrow-pushing diagram where the interaction is shown by a group of directed half arrows from one reactive MO to the other reactive MO.

Since arrow-pushing mechanism $A$, uniquely defines the single transition state (see [25]), by knowing the atom mapped $R$, atom mapped $P$, and $A$, OrbChain can uniquely determine the reactive pair of orbitals, $(m_1^{(*)}, m_2^{(*)})$, in $R$. Also, by knowing the atom mapped $R$ and two orbital pairs (assumed to be reactive) $(m_1^{(*)}, m_2^{(*)})$, OrbChain can uniquely determine the atom mapped $P$ and $A$.

We summarize the OrbChain in the Equation 1.

$$\text{OrbChain}: \begin{cases} (1) \ R, P \xrightarrow{\ A\ } (m_1, \ m_2): \text{ Atom mapped } R, P \text{ and } A, \text{ determine the reactive MOs.} \\ (2) \ R \xrightarrow{(m_i, \ m_j)} P', A': \text{ A mechanism is an interaction between a pair of MOs } m_i, m_j. \end{cases}$$
(1)

The problem of predicting the outcome of a radical reaction mechanism is to find the $P$ and $A$ given $R$. Using OrbChain, this problem is equivalent to finding the reactive pair of molecular orbitals $(m_1^{(*)}, m_2^{(*)})$ given $R$.

To determine $P$ and $A$ in accordance with the model described above, the first step is for OrbChain to identify all molecular orbitals (MOs) by performing an iterative process over the set of nodes $a_j^i$ present in the reactant molecules $R_i$. Each MO is associated with four distinct parameters $m = (a, e, n, c)$, where $a$ represents the atom corresponding to the MO (i.e., the central atom of the MO), $e$ denotes the number of electrons involved in the MO (0, 1, or 2), $n$ indicates the atom adjacent to the atom $a$ in the case of a bond orbital (such as a $\pi$ or $\sigma$ bond), and $c$ signifies the possible chain of filled or unfilled MOs (such as a $\pi$ system). By utilizing $c$, we can uniquely determine $P$ and $A$ by recursively following the electron transfers from one MO to the next.

In a reactant molecule $R_i$ with $n_i$ atoms, we show the set of found MOs as $m_i = \{m_i^j\}$. Then the total number of possible (not plausible) mechanistic radical reactions from the reactant molecules $R = \{R_i\}_{i=1}^{n_r}$, is approximated in Equation 2. This would be a large number for molecular systems with approximately 20 atoms. Within this vast space of possible mechanistic reactions, the number of plausible ones (e.g., $(m_1^{(*)}, m_2^{(*)})$) is tiny, usually less than or equal to three.

$$|X| \approx \binom{\sum_{i=1}^{n_r} |\{m_i\}|)}{2}$$
(2)

## 4.2 Two-Step Prediction

To address the challenges associated with the multitude of potential mechanistic reactions, we adopt a method inspired by [20]. We begin by shrinking the pool of potential mechanisms through a filtering process called reactive sites identification. In this step, we filter the potentially reactive MOs, so we can only consider the interaction between the potentially reactive MOs. This filtering process offers two key advantages: (1) reduced computation time and (2) removal of false positives, leading to improved precision and recall in the overall prediction. After reactive sites identification, we employ a reaction ranker, a machine learning model that ranks the possible interactions $(m_i, m_j)$ within the refined space of filtered MOs. The reaction ranker facilitates the identification of the most favorable orbital interactions for the specific set of reactants under examination.

### 4.2.1 Reactive Sites Identification

We divide orbitals into reactive and non-reactive orbitals based on their role in the reaction. Based on statement (2) in Equation 1, there are only two reactive orbitals for each radical mechanistic step $(R, P, A)$. Instead of predicting the reactive MOs ($a = atom, e, n, c$), it is more convenient to predict the label of atom $a$, which is associated with the MO as follows:

$$g(a_j^i) = \begin{cases} 0 & m^* \notin \{m \mid m = (a_j^i, e, t, c)\} \\ 1 & m^* \in \{m \mid m = (a_j^i, e, t, c)\} \end{cases}$$
(3)

Then, finding a set of potentially reactive atoms in a group of reactant molecules is a classic node classification problem in the context of graph learning.

We adopt the reactive sites identification method from [20], which represents atoms using continuous vectors based on predefined atomic and graph-topological features. We expand these atom descriptors by incorporating more atomic features such as the number of radical electrons. For all the atoms within the RMechDB datasets, we extract these feature vectors to train a classifier that can classify each atom into reactive or non-reactive. The parameters of the trained model and the statics of the training data can be found in the Appendix.

To improve this atom classification and circumvent the manual feature extraction part, we deploy a form of graph convolution neural network (GNN) suitable for the atom classification [30; 5; 17] augmented with attention mechanisms [31]. More information on the parameters and specifications of the GNN can be found in the Appendix. Table 2 shows the statistics of the training data for the reactive sites identification part.

The results of reactive sites identification are presented in Table 2. While these methods achieve reasonable accuracies, they overlook a crucial aspect: the context of the reaction, which encompasses both spectator and reagent molecules. Given that the reactivity of various sites and functional groups can be significantly influenced by the reaction context, we will propose context-aware approaches in Sections 4.3 and 4.4 to address this limitation.

### 4.2.2 Plausibility Ranking

Once reactive sites are identified within a set of reactant molecules, we employ OrbChain's statement (2) for all combinations of reactive sites. Since each reactive sites (i.e., atoms) corresponds to an MO, this process yields all possible mechanistic reactions from the set of input reactants. Each of these mechanistic reactions includes atom-mapped products and arrow codes representing interactions between predicted MOs. To rank the proposed mechanistic reactions, we utilize a deep Siamese neural network [32] designed for entity ranking. During training, this network takes a pair of plausible and implausible mechanistic reactions as input and produces a real-valued number for each. The network's parameters are learned by minimizing the following loss function:

$$\mathcal{L} = \sigma[f(\text{Rxn}_{plausible}) - f(\text{Rxn}_{implausible})] \tag{4}$$

In the given loss function, $f$ represents a neural network, $\sigma$ denotes the logistic function, and Rxn represents a mechanistic reaction. Minimizing this loss function allows the function $f$ to assign higher scores to plausible reactions. To establish a relative plausibility score, we train the model using pairs of $\text{Rxn}_{plausible}$ and $\text{Rxn}_{implausible}$ with identical reactants. During training, the $\text{Rxn}_{plausible}$ is a sample of the RMechDB dataset while the $\text{Rxn}_{implausible}$ is generated using OrbChain's statement (2) with randomly chosen non-reactive MOs of the same sample from RMechDB. Once the model is trained, the output of function $f$ can be interpreted as the plausibility score. We use this output to rank all generated mechanistic steps from the combination of reactive sites.

The performance of the model is significantly influenced by the choice of reaction representation and the corresponding neural network architecture for function $f$. We explore and compare four different reaction representations and their corresponding neural architectures: (1) pre-defined feature vectors [20], (2) reactionFP [33] with three different molecular fingerprints including atom pair fingerprints (AP) [34], Morgan fingerprints [35], and Topological Torsion fingerprints (TT) [36], (3) Differential Reaction Fingerprint (DRFP) [37], and *rxnfp*, a text-based reaction representation based on a pre-trained model for yield prediction [38]. Detailed descriptions of the methods for generating implausible reactions, the specific parameters used for each neural network, and the parameters used for each reaction fingerprint can be found in the Appendix. The comparative results of the plausibility ranking are presented in Table 3.

### 4.3 Contrastive Learning

As stated above, the context of a reaction can affect the dynamic of orbital interactions by changing the reactivity of different functional groups. An informative atom representation for reactive sites identification must take this context into account. In this section, we solve this problem by proposing a new method similar to [14] which computes and learns the atom representations by considering the entire context of the reaction. The key idea is to consider a pair of atoms and predict the most

reactive pair instead of predicting the reactive atoms separately. We take into account the full context of the reactions by considering all possible atom pairs and comparing them with the most reactive one in a contrastive learning manner. This contrastive learning model can approximate the probability distribution in Equation 5. Notably, this method not only captures the impact of the reaction context but also streamlines the prediction process into a single step, resulting in faster inference when compared to the two-step prediction method.

$$\mathbb{P}\Big((m_i, m_j) = (m_1^*, m_2^*) \mid R\Big) \tag{5}$$

### 4.3.1 Atom Pairs and Atom Descriptor

First, we establish a baseline for approximating the probability mentioned above using atom descriptors. In this approach, the positive data consists of the most productive reactions $(R, P, A)$(i.e., each sample of the RMechDB dataset), while the negative data includes all other possible reactions from the same set of reactants $(R, P', A')$. To train the contrastive model, reactions, whether positive or negative are represented as a pair of atoms where each atom is the representative of its reactive MO. The targets $y_{ij}$ for an atom pair $(a_i, a_j)$ is obtained as shown in Equation 3. We calculate the marginalized probability (Eqn. 5) by considering all possible atom pairs in reactants $R$. This approach has two advantages: (1) It enables one-step reaction prediction by identifying the most reactive pair of MOs, determining the product and arrows according to Orbchain (statement (2)). (2) It reduces false negatives by not discouraging less reactive, yet still plausible, MO pairs. These pairs are ranked highly, with the most reactive MO pair receiving the highest ranking.

$$y_{(a_i, a_j)} = \begin{cases} 1 & m_1^* = (a_i, e, n, c) \;\; \& \;\; m_2^* = (a_j, e', n', c') \\ 0 & \text{Otherwise}; \end{cases} \tag{6}$$

Figure 2 (left side) shows the schematics of the constrastive model. Table 4 presents the results of this method, while the Appendix provides details on the objective function, parameters of the contrastive model (Figure 2), and the atom descriptor.

### 4.3.2 Rxn-Hypergraph

Considering all atom pairs in a contrastive learning fashion would take the reaction context into account. However, we can compute a more informative representation of atom pairs by utilizing the Rxn-Hypergraph. Rxn-Hypergraph introduced in [27] is a graph attentional neural network that operates on the hypergraph of the entire reaction. Instead of using the atom descriptor, we can train a Rxn-Hypergraph model, to automatically learn a contextual representation of all atoms of the reactant sides.

To adapt the Rxn-Hypergraph for the reaction predictor task (OrbChain Statement (2)), we modify the original structure of the hypergraph by duplicating the reactants on both sides of the graph. This modification allows us to provide the reactants on one side while maintaining the full context on the other side. Following the training procedure of [27], we compute atom representations and generate pairs of atoms. These pairs are then fed into the contrastive network depicted in Figure 2 to obtain a ranked list of atom pairs. Each atom pair corresponds to an interaction between two orbitals, which enables us to generate products and arrows using OrbChain. The results of the reaction predictor using atom pair prediction with Rxn-Hypergraph are presented in Table 4.

### 4.4 Text Representation and Sequence to Sequence Models

Considering the SMILES string as the text representation of molecules [39; 40], a chemical reaction can be seen as the transformation of a sequence of characters (reactants) to another (products). This makes the sequence-to-sequence models, such as the Transformer [41; 42] and models based on the recurrent neural network architecture, a suitable predictive model for chemical reaction predictions [43; 44; 45; 28], retrosynthesis prediction [46] and molecular optimization [47].

Existing text-based models for chemical reaction prediction exhibit limitations in various aspects, notably in terms of interpretability and balance in their predictions. SMILES representations are not unique and do not encode inherent molecular properties, such as invariance to atom permutations. Consequently, these models often require extensive data augmentation to improve performance. Nonetheless, we seek to leverage the success of these models and apply them to the prediction of

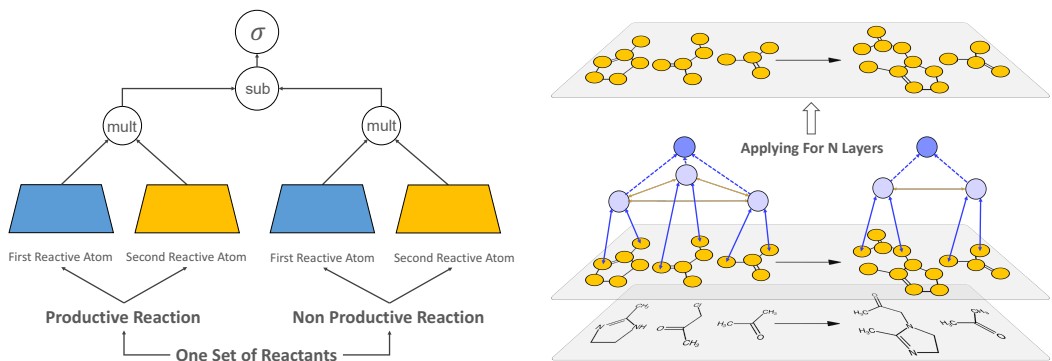

Figure 2: Left: The architecture of the contrastive learning approach. Right: The schematic depiction of the Rxn-hypergraph.

radical mechanistic reactions. In particular, we adopt the pioneering text-based reaction predictor, Molecular Transformer [28], which utilizes a bidirectional encoder and autoregressive decoder with a fully connected network for generating probability distributions over possible tokens. The pre-trained Molecular Transformers were trained using different variations of the USPTO dataset [16]. During training the encoder computes a contextual vector representation of the reactants by performing self-attention on the masked and randomly augmented (non-canonicalized) SMILES string of the reactant molecules. The decoder then uses the encoder output and the right-shifted SMILES string of the products to autoregressively generate the product tokens. Since the radical reactions in RMechDB are not labeled with reactants and reagents, we used the model which was pre-trained using the USPTO_MIT_mixed dataset [16; 48; 49].

### 4.5 Fine-tuning Using RMechDB

Molecular Transformer enables the fine-tuning of pre-trained models for downstream tasks like radical reaction prediction. In our approach, we utilize pre-trained models and conduct reactant-to-product sequence translation. During the fine-tuning process, our only augmentation technique involved rearranging the reactant molecules within the SMILES string. Specifically, for each reaction containing $N$ reactant molecules, we employed $N$ SMILES strings with reactants randomly reordered. We removed all the atom mappings and arrow codes from the RMechDB training data and fine-tuned the model using the augmented training set of the RMechDB.

Table 4 shows the performance of the text-based prediction for both pre-trained and fine-tuned versions of the Molecular Transformer. We also include detailed information on training and fine-tuning parameters, as well as tokenization statistics.

## 5 Results and Discussion

### 5.1 Performance on RMechDB

We assess the performance of the two-step prediction method, comprising reactive sites identification and plausibility ranking. The top $N$ accuracy of the reactive sites identification on the combined test datasets of RMechDB are presented in Table 2. We observe that GNN models outperform the method based on the atom descriptor (predefined feature extraction). This behavior is expected as the atom descriptor is limited to a certain radius around the atom (in this case the radius is set to three). However, the number of GNN layers can be optimized to construct the most informative atom representations. The advantage of GNNs is more evident for the atmospheric test data where there are usually more molecules present in the context of the reaction.

The second model of the two-step prediction is the Siamese architecture that ranks the reactions based on their chemical plausibility. Four different architectures were used for different reaction representations: Pre-defined feature vectors from [20], reactionFP, DRFP, and the *rxnfp*. The results of the top $N$ accuracy for the combined test sets of RMechDB are shown in Table 3. The table shows that DRFP outperforms the reactionFP and the feature extraction methods. We believe the reason is

mainly because of the different nature and underlying chemistry of the radical mechanistic reaction and the USPTO reaction, which was used to pre-train the *rxnfp*.

Table 2: The performance of different methods for reactive sites identification. Each number represents the percentage of reactions for which both reactive atoms are identified within the top $N$ predictions.

| Method | Top2 | Top3 | Top5 | Top10 |
|---|---|---|---|---|
| Atom Descriptor | 75.1 | 81.5 | 89.3 | 96.7 |
| GNN | 76.9 | 83.6 | 92.1 | 97.9 |

To predict the outcome of a mechanistic radical reaction using the two-step method, we must perform both reactive sites identification and reaction ranking. Given the performance of each individual predictor, the best combination would be the GNN and DRFP. Therefore in Table 4, we use this combination to compare the performance of the other two methods with the two-step prediction.

Table 3: The performance of different methods for plausibility ranking. Each number represents the percentage of reactions for which the correct mechanism is predicted in the top $N$ plausible reactions.

| Method | | Top1 | Top3 | Top5 | Top10 |
|---|---|---|---|---|---|
| Pre-defined Feature Vectors | | 73.1 | 79.2 | 88.3 | 96.3 |
| | AP | 74.6 | 82.3 | 90.2 | 97.8 |
| reactionFP | Morgan2 | 74.3 | 81.9 | 90.0 | 97.3 |
| | TT | 74.3 | 82.4 | 90.0 | 97.8 |
| DRFP | | 78.6 | 90.2 | 95.1 | 100.0 |
| *rxnfp* (pretrained) | | 75.9 | 86.2 | 94.3 | 97.9 |

As it is shown in Table 4, the contrastive learning approach yields the most accurate prediction across all the metrics. In terms of inference time, the contrastive learning approach is faster than the two-step prediction as it only consists of one neural network. This becomes crucial in the case of pathway search where we exponentially expand the tree of the mechanistic pathways by predictions. However, the advantage of using the two-step method becomes more evident when the size of the reactant molecules increases. In Figure 3, we show the performance of all three methods based on the number of heavy atoms on the reactant sides. As expected, the prediction of both the contrastive method and text-based models becomes more faulty when big molecules react. This can be explained using the fact that the reactive sites identification part of the two-step method is mainly dependent on a local neighborhood of atoms and is not significantly affected by the size of the molecules.

The text-based models are outperformed by the other two graph-based methods. Although they offer faster inference times, they yield less accurate predictions. This poor performance is mainly because Molecular Transformer model is trained on the USPTO_MIT_mixed dataset. Therefore, it learned to predict the only major product of overall transformations that are mostly polar. On the contrary, RMechDB data are balanced, mechanistic, and involve radical species. Align with this low performance, according to Table 4, fine-tuning on the relatively small RMechDB dataset results in a slight decrease in the performance. This can be attributed to the dissimilarities between RMechDB and the USPTO dataset. RMechDB comprises intermediate products of transformations and includes radical reactions with compounds not found in the USPTO dataset. During the tokenization process of the RMechDB reactions, new tokens emerge that have not been extracted from the USPTO dataset.

We also provided a separate performance of the predictors with respect to different radical reaction categories provided in RMechDB.

## 5.2 Pathway Search

After successfully predicting the outcomes of mechanistic radical reactions, we can chain these predictions to construct mechanistic pathways. Starting from a set of reactant molecules, we perform a series of predictions by using each of the top $N$ predicted products as the reactants for the next prediction. This would form a tree structure with the starting reactants as the root. This tree can be expanded to a desired depth, representing numerous mechanistic pathways leading to different

Table 4: top $N$ accuracy of all the three reaction prediction models on Core and Atmospheric test sets of RMechDB. For the text-based models, a prediction is considered to be correct, when at least one of the non-spectator product molecules is predicted correctly. For the text-based models, we use (p) and (f) for pre-trained, and fine-tuned models respectively.

| Model | Variant | Core | | | | Atmospheric | | | | Time |
|---|---|---|---|---|---|---|---|---|---|---|
| | | top $N$ | | | | top $N$ | | | | |
| | | 1 | 2 | 5 | 10 | 1 | 2 | 5 | 10 | |
| Two-step | Best Combination | 62.4 | 71.9 | 93.2 | 97.2 | 60.4 | 70.9 | 91.6 | 96.3 | 1.38 |
| Contrastive | Atom Descriptor | 62.9 | 73.8 | 94.2 | 96.5 | 61.0 | 73.6 | 93.0 | 94.4 | 0.08 |
| | RxnHypergraph | 64.3 | 74.1 | 95.1 | 97.4 | 62.1 | 74.8 | 94.1 | 95.9 | 1.45 |
| Text-based | Pretrained | 58.2 | 64.3 | 84.2 | 91.0 | 58.0 | 67.3 | 82.6 | 91.0 | 1.30 |
| | Fine-tuned | 57.7 | 64.0 | 83.9 | 90.4 | 57.1 | 66.8 | 82.2 | 90.3 | 1.30 |

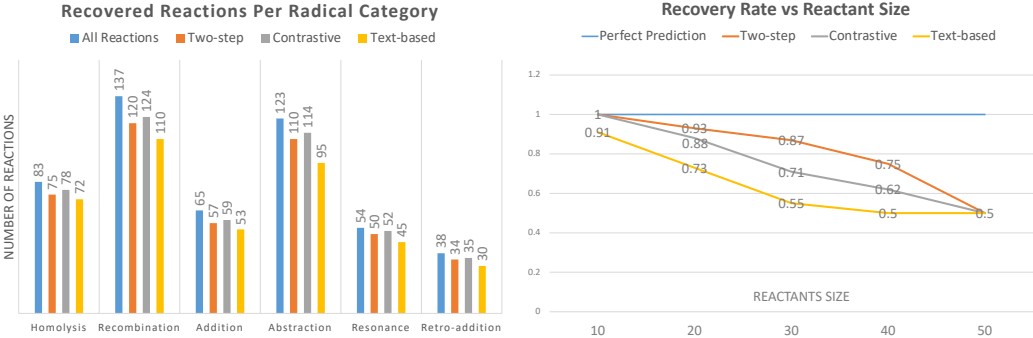

Figure 3: Left: The number of recovered reactions in the top 5 for different methods and different classes of RMechDB reactions. Right: The recovery rate of different methods with respect to the size of the reactants.

products. These pathways are crucial for identifying synthetic routes, exploring intermediate products, and aiding in mass spectrometry analysis with broad applications to various fields, including drug design and drug degradation.

We create a dataset consisting of 100 radical reactants paired with target molecules that are expected to be formed from the reaction. Each pair is accompanied by a specified depth, indicating the length of the mechanistic chain reactions required to reach the target. We generate this dataset by simulating atmospheric conditions, taking inspiration from the atmospheric reactions observed in the RMechDB dataset and atmospheric literature [50]. The reactants in our dataset primarily consist of Isoprene, a prevalent atmospheric compound, along with other atmospheric molecules like radical Oxygen and Hydroxyl radical. To find these targets, we expand and search the tree of the mechanistic pathways by employing a breadth-first search algorithm [51]. To expand the tree for each of the 100 reactants, we use the top 10 predictions at each step (i.e., the breadth of 10 to expand the tree), and we expand the tree up to the given depth. Given the average inference time of the predictive models presented in Table 4, we used the fastest method with interpretable predictions which is the contrastive model with atom descriptor. Upon running this pathway search for these 100 pathways, we observed a significant recovery rate of 60% meaning that for the 60% of the reactants, the given target was found within the expanded tree. These pathways along with their targets and detailed information on the pathway search are presented in the Appendix.

### 5.3  RMechRP: Online Reaction Predictor

We developed RMechRP as an online tool for mechanistic radical reaction prediction. RMechRP, which stands for **R**adical **Mech**anistic **R**eaction **P**redictor, is accessible at `http://deeprxn.ics.uci.edu/rmechrp/`. It offers two interfaces: (1) Single-step Predictor; and (2) Pathway Predictor. These interfaces are equipped with the highest performing combination of the two-step prediction

models and the contrastive learning model respectively to offer the best blend of interpretability, accuracy, and fast inference. The Single-Step Predictor interface, accessible at `http://deeprxn.ics.uci.edu/rmechrp/singlestep`, provides users with the capability to input a set of reactant molecules. By specifying a few parameters, such as the number of reactive atoms, the system utilizes the best combination of the two-step prediction model described above to identify all possible radical mechanistic steps. Then all the predicted reactions will be ranked and displayed with side information, including arrow codes, SMIRKS, reactive orbitals, and plausibility scores. The Pathway Predictor interface, accessible at `http://deeprxn.ics.uci.edu/rmechrp/pathway`, is designed for searching a specific target molecule(s) within the tree of mechanistic pathways rooted at the input reactants. Users have the option to input a set of reactants and specify a set of target molecules, which can be provided in the form of molecular structures or molecular masses. Parameters such as the depth and breadth of the mechanistic pathway tree can be configured. The system then employs the contrastive learning model described above to expand and explore the mechanistic pathway tree. Upon finding the target molecule within the expanded tree, the system will present synthetic pathways from the initial reactants to the target molecule or molecules. More details on both interfaces and how to use them are presented in the Appendix.

## 6 Conclusion

We developed and compared multiple radical reaction predictors operating at the mechanistic level using the RMechDB database of radical elementary reaction as the source of training and evaluation data. We show that the contrastive learning approach incorporated with the graph representation of chemical structures would yield the highest accuracy. The models that merely build upon the text representation of chemical structures require a larger dataset to achieve the same level of accuracy. We show that our approach to reaction prediction is capable of providing chemical interpretability down to the interaction of molecular orbitals. It also offers pathway interpretability by breaking down an overall transformation into balanced pathways of elementary steps. Considering these chemically balanced pathways would identify interesting byproducts that can potentially give rise to pathways that are usually ignored in representing overall transformation. Additionally, we developed online interfaces for performing single-step and pathway prediction of radical reactions. Our radical reaction predictor is publicly available through an online interface at `http://deeprxn.ics.uci.edu/rmechrp`.

Finally, it is important to identify the limitations of the current method and note that the radical reaction predictor can be further extended in several aspects. First, although RMechDB provides high-quality mechanistic reaction data, it only includes 5500 mechanistic reactions. Developing purely data-driven machine learning models such as LLMs, with minimum dependency on physical and chemical information requires much larger datasets. Additionally, RMechDB is focused on textbook reactions and atmospheric reactions. This focus may restrict the applicability of the proposed predictor to other types of radical reactions, such as polymerization processes. In the context of pathway search and the expansion of the search tree, we employed a straightforward breadth-first search algorithm. This algorithm allows us to expand and search the entire tree of pathways, however, the exponentially growing computations would impede the search process. An interesting improvement could involve the adoption of more sophisticated search methods, like those mentioned in [52], which are not only faster but also capable of exploring a significant portion of the search space.

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

# 7 Appendix

In this appendix, we provide a comprehensive description of the experimental details and environments in which the experiments were conducted. Additionally, we present detailed information and data pertaining to the pathway search. Furthermore, we offer an explanation of the various interfaces of the RMechRP software, which serves as the pioneering online radical reaction predictor. Each section in this appendix corresponds to the section with the same title in the main article. Finally, all the experiments are conducted using a single NVidia Titan X GPU.

## 7.1 Reactive Sites Identification

For the Atom Fingerprint model, we constructed a fingerprint of length 800 for each atom. This fingerprint includes 700 graph topological features explained in [20] and 85 atomic features including a one-hot vector for atom type, and chemical features of the atoms such as valance and electronegativity. The graph topological features are extracted using a neighborhood of size three. The extracted fingerprints are fed into a fully connected model with an output layer for binary classification. For the GNN model, we used the atomic feature for the initial representations of atoms. The model consists of four GNN layers with an output layer for binary classification.

Combining both training sets presented in RMechDB [25], we extracted over 51000 atoms to train each of the models above. Both models are evaluated using a combination of two test sets in RMechDB and the top $N$ accuracy of models is reported in Table 2 of the main article. Table 5 represents the parameters used for training the models.

Table 5: The parameters used for training the models for reactive sites identification.

| Model | Batch Size | Num Layers | Layers Dim | Act | Reg | Num Att Heads |
|---|---|---|---|---|---|---|
| Atom Fingerprint | 32 | 3 | 512-256-1 | GELU | $L_2$(5e-5) | - |
| GNN | 32 | 4 | 64-64-64-1 | ReLU | Dropout (0.3) | 6 |

## 7.2 Plausibility Ranking

For the plausibility ranking experiments, we used the following four methods for representing chemical reactions:

**Feature Extraction**: We use the same features explained in [20] which results in extracting a vector of length 3200 for each reaction.

**reaction*fp***: We use the RDKit [53] implementation of reaction*fp* [33]. For all three fingerprint types (Atom Pair, Morgan2, and Topological Torsions), we use a fingerprint of size 2048, with a bit ratio of 0.2. We considered nonagent molecules with a weight of 0.4 and agent molecules with a weight of 1.0.

**DRFP**: We use the DRFP fingerprint [37] with a size of 2048 with a min and max radius of zero and four, while including the hydrogen atoms and rings.

***rxnfp***: We use the default tokenizer and pretrained model for the *rxnfp* [38] which results in fingerprints of length 256.

For training, we use a combination of both training sets in RMechDB. For each sample of the training data (productive reaction), we generate (at most) 40 negative samples (unproductive reactions) by randomly sampling molecular orbitals other than the reactive MOs $(m_1^*, m_2^*)$. This results in a data set of over 185000 pairs of productive and unproductive reactions. To train the plausibility rankers for each method, we use the parameters explained in Table 6.

## 7.3 Atom Pairs and Atom Descriptor

For the contrastive learning method using atom descriptors, we use the same atomic feature and graph topological features above to represent one single atom. Specifically, for the graph topological features, we use the neighborhood of size one. These features plus the atomic features result in

Table 6: The parameters used for training the models for the plausibility ranking.

| Model | Batch Size | Num Layers | Layers Dim | Act | Reg |
|---|---|---|---|---|---|
| Feature Extraction | 32 | 3 | 512-256-1 | GELU | Dropout (0.5) |
| reaction*fp* | 32 | 3 | 400-200-1 | GELU | Dropout (0.5) |
| DRFP | 32 | 3 | 400-200-1 | GELU | Dropout (0.5) |
| *rxnfp* | 64 | 2 | 128-1 | GELU | Dropout (0.5) |

a vector of length 140 for atom representation. Using these vectors, we train a contrastive model depicted in Figure 2 (left) of the main article. The objective function to train this contrastive model is as follows:

$$\mathcal{L} = 1 - \sigma\Big([f(a_1^*) \times g(a_2^*)] - [f(a_1') \times g(a_2')]\Big) \tag{7}$$

$$Score = \sigma([f(a_1) \times g(a_2)]) \tag{8}$$

Where $a_1^*$ and $a_2^*$ are the atoms of the reactive MOs $m_1^*$ and $m_2^*$, while $a_i'$ are randomly chosen atoms. Both $f$ and $g$ functions are characterized by a fully connected neural network. The first reactive atoms in both productive and unproductive reactions are fed through the same network $f$, and similarly, the second reactive atoms are fed through the same network $g$. The outputs of both $f$ and $g$ are single real-valued numbers, which, when multiplied together, yield a score for the respective reaction. These scores are then utilized to construct the objective function, aiming to maximize the score of the productive reaction compared to the unproductive reactions using the same reactant set. Once the model is trained, the normalized score (Equation 8) can be interpreted as the reactivity of an atom pair.

We use a combination of both training sets in RMechDB to train $f$ and $g$. For each productive reaction, we form unproductive reactions by considering at most 15 samples of $(a_1', a_2^*)$, $(a_1^*, a_2')$, and $(a_1', a_2')$. This negative sampling results in a dataset of over 200000 pairs of productive and unproductive atom pairs. We use this training dataset to minimize the objective function 7.

Both $f$ and $g$ have similar architectures that consist of three fully connected layers with a GELU activation function and a dropout with a rate of 0.5 applied to all layers. The dimensions of the layers are 128, 64, 1.

## 7.4 Rxn-Hypergraph

We use the Rxn-Hypergraph to replace form atom descriptors that are extracted automatically for minimizing the objective function 7. After processing the Rxn-hypergraph for N layers, the generated atom descriptors are used in the same setting above for the same minimization objective. Here in Table 7 we describe the parameters we use for training the Rxn-Hypergraph.

## 7.5 Text Representation and Sequence to Sequence Models

In order to develop a text-based radical reaction predictor, we utilize the pre-trained molecular transformer which was trained using the USPTO_MIT_mixed dataset. We also used the tokenizer developed by the molecular transformer. This tokenizer yields 523 distinct tokens for the USPTO_MIT_mixed dataset. There are nine tokens from the RMechDB dataset that do not match the 573 tokens of the USPTO. Therefore, we used the *"unknown token"* to represent these nine tokens.

For fine-tuning the pre-trained model, we used the combination of both RMechDB training sets. We fine-tune the model using a simple data augmentation described in Section 4.5 for 10 epochs. Finally, for the evaluation of the text-based models, we considered all the generated *"unknown token"* as correct tokens.

Table 7: The parameters used for training the Rxn-Hypergraph for the contrastive model.

| Batch Size | Num Layers | Layers Dim | Act | Reg | Num Att Heads | Learning Rate |
|---|---|---|---|---|---|---|
| 32 | 5 | all 64 | GELU | $L_2$(5e-5) | 6 | 0.001 |

## 7.6 Pathway Search

In the Pathway Search section, we conducted an experiment involving the execution of the pathway search for a set of specific reactants. Each of these reactants was associated with a desired target molecule, which was expected to be found within the mechanistic pathway tree. Additionally, a set of distinct parameters such as the context, depth, and breadth is associated with each reactant.

To provide detailed information and facilitate reproducibility, we have included supplementary materials accompanying this work. The full list of these reactants used for the pathway search is accessible at `http://deeprxn.ics.uci.edu/rmechrp/pathway`. This list contains the reactants, corresponding targets, the provided context (if any), and the anticipated depth at which the target molecule is expected to appear within the mechanistic pathway tree. For each entry of this list, we include a visualization of the identified pathways leading to the specified target molecules. It presents the discovered pathways that were found during the experiment. These materials serve to provide comprehensive insights into the pathway search process and its outcomes, enabling readers to reproduce and further explore the obtained results.

## 7.7 RMechRP: Online Reaction Predictor

In addition to the methods and results presented in the main article, we have developed an online web server that enables users to utilize the trained models for predicting the outcomes of mechanistic radical reactions with the highest levels of interpretability of the outcome. RMechRP (**R**adical **Mech**anistic **R**eaction **P**redictor) accessible via the anonymized link `http://deeprxn.ics.uci.edu/rmechrp`. RMechRP offers two interfaces: Single-step prediction and Pathway search.

**Single-Step Predictor** predicts the outcome of a mechanistic reaction with a single transition state. Users have the option to either input the reactants in written form or draw them using a drawing tool provided on the web server. Additionally, users can specify the reaction conditions, with the current option being standard temperature and pressure. The number of reactive molecular orbitals (MOs) to be considered can also be specified by the user.

To ensure flexibility, users can choose to filter out reactions that violate specific chemical rules, such as Bredt's rule [54]. Once the input and conditions are set, the user can click the predict button. The system will then run the two-step prediction model, as described above, to generate and rank the potential products. These predicted products will be displayed, accompanied by additional information such as arrow codes, reactive MOs, and the mass of the products. The single-step predictor is accessible via the anonymized link `http://deeprxn.ics.uci.edu/rmechrp/singlestep`. Figure 4 shows the single-step interface and the displayed predictions for a simple reaction.

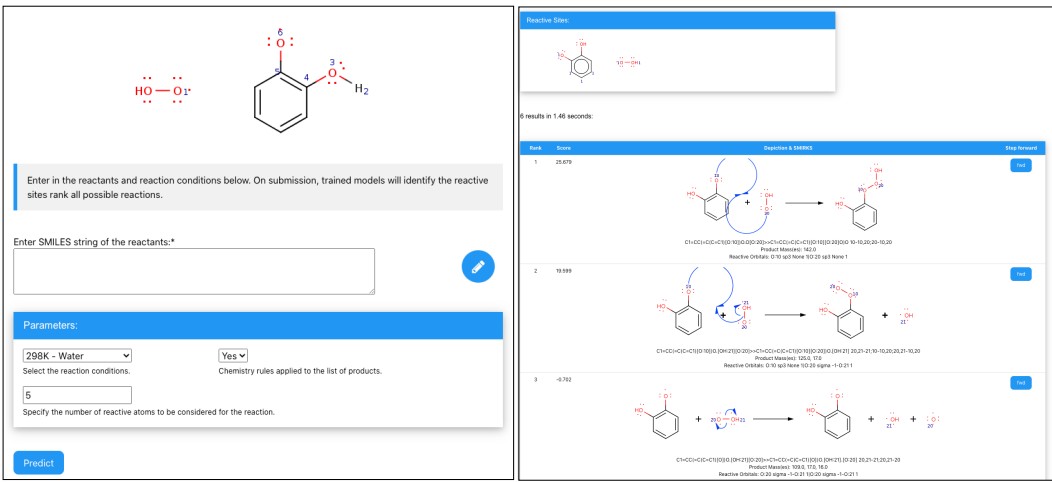

Figure 4: The single-step interface with the predictions of a simple reaction. Left: the input panel. Right: the table displaying the ranked predictions.

**Pathway Predictor** forms the tree of the mechanistic pathways up to a given depth and breadth. Users have the option to either input the reactants in written form or draw them using a drawing tool provided on the web server. Users must also input a set of targets (either mass or chemical structure) to look for within the expanded tree of the mechanistic pathways. users have the ability to provide a context for the reactions. The context consists of a set of molecules along with their corresponding frequencies of appearance within the mechanistic pathway tree. When a molecule from the context is consumed in a reaction, the system can automatically reintroduce that molecule back into the pathway tree. The frequency of appearance indicates how many times a molecule can be added to the mechanistic pathway tree.

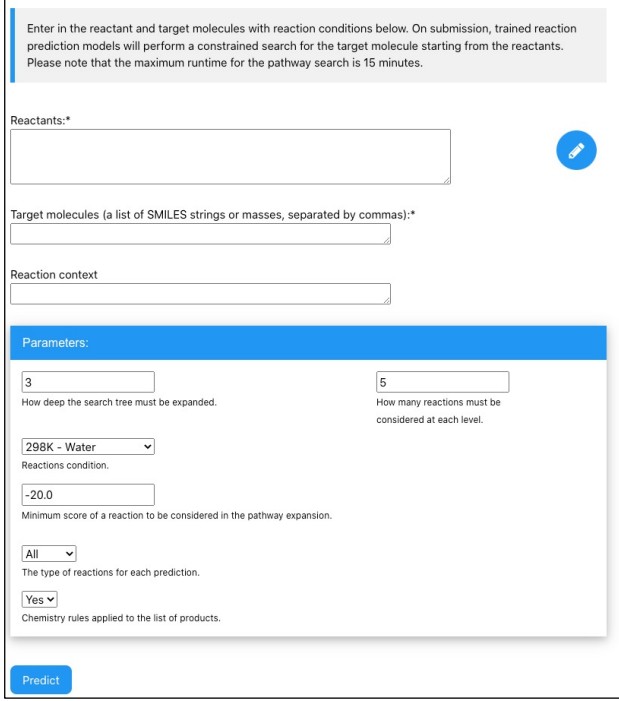

Figure 5: The pathway search interface.

In addition to the context, there are several additional parameters that can be specified by the user. These parameters include:

Depth of Pathway Search: Users can define the depth of the pathway search, which determines how many reaction steps will be explored in the mechanistic pathway tree.

Breadth (Branching Factor) of Pathway Search: This parameter controls the branching factor of the pathway search, influencing the number of alternative reaction pathways that will be considered.

Application of Chemistry Rules: Users have the option to apply certain chemistry rules during the pathway search. These rules can be used to filter out reactions that violate specific chemical principles or constraints.

Score Threshold: Users can set a threshold value to consider only reactions with scores higher than the specified threshold. This helps narrow down the focus to more favorable or promising reactions.

These additional parameters allow users to customize their pathway search and refine the results based on their specific requirements and preferences. By leveraging these features, users can gain deeper insights into the mechanistic pathways and explore a wider range of possible reaction outcomes. The pathway search interface is accessible via the anonymized link `http://deeprxn.ics.uci.edu/rmechrp/pathway`. Figure 5 shows the pathway interface and the required parameters.

