# OpenReview forum: "AI for Interpretable Chemistry: Predicting Radical Mechanistic Pathways via Contrastive Learning"
_NeurIPS.cc/2023/Conference — NeurIPS 2023 poster_

### Official Review · Reviewer_G4db · 2023-06-27

**Soundness:** 3 good
**Presentation:** 2 fair
**Contribution:** 3 good
**Rating:** 5
**Confidence:** 4

**Summary:**

The paper proposes a set of methods for modeling chemical reactions that involve radicals during the reaction process. The authors first introduce the current landscape of chemical reaction datasets based primarily on the USPTO and discusses the USPTO shortcomings in terms of reaction interpretability and its inability to showcase reaction involving multiple steps. Next the authors briefly describe the RMechDB dataset, which does contain pathways with radicals, followed by a description of their methods. The methods are based on OrbChain, which provides a standardized way of describing chemical reactions with radical and arrow pathways. After that, the authors introduce their predictive methods, including two-step prediction, plausibility ranking, contrastive learning with a reaction hypergraph and text-based sequence to sequence models. The authors conduct experiments for all the aforementioned methods to further understand their capability in accurately describing radical reaction pathways on the RMechDB dataset. The performances of the different methods vary across different settings of the conducted experiments. The authors then provide a pathway search example, further description of their package and a conclusion.

**Strengths:**

The paper provides has the following strengths:
* Originality: The papers provides a new perspective on chemical reaction modeling that involves radicals and is also more interpretable for classically trained chemists.
* Quality: The paper describes and analyzes four different and relevant methods for the radical modeling problem and provides clear motivations for their importance.
* Clarity: The paper motivates the problem they address quite and describe the necessary background.
* Significance: Expanding the capabilities of machine learning models to provide more interpretable reaction models with more steps in the reaction process could have significant impact on various chemistry related problems.

**Weaknesses:**

The paper could be further improved:
* Providing a clearer description of the context of the results related to original problem the authors motivated. How well do the described methods provide more interpretability to chemical reaction modeling? How do the metrics the authors measure relate to that original premise? [quality, clarity, siginificance]
* The authors only briefly describe pathway search, but provide little context for what their results mean. What does a recovery rate of 60% imply? How does the reaction tree look like and how interpretable is it? [clarity]
* The authors refer the reader to the appendix very often, which I think contains a lot of significant information needed to fully understand the experimental results. I recommend putting more of that information in the main paper. [quality, clarity significance]
* The authors only provide a brief description of the RMechDB dataset and its unclear if that paper had any modeling methods the authors could compare their proposed methods to. Further clarification on this would be helpful. [clarity]

**Questions:**

* Could you clarify if the RMechDB paper provided any modeling methods?
* Would it be possible to provide more context for the results, ideally in the figure and tables themselves, to further understand the experiments? How good does a Top1, Top2, Top5 score need to be practically useful, for example?
* Is it possible to have text-based methods also express the intermediate steps in reactions involving radicals? Why or why not?

**Limitations:**

The authors do not provide a detailed discussion on limitations. A discussion on limitations would make the paper stronger.

---

> ### Author Rebuttal · Authors · 2023-08-10
>
> We appreciate the constructive comments from the reviewer. We have addressed most of their comments in the revised version of the manuscript.
>
> **Comments on weaknesses**
> * We agree that the description of the pathway search experiment is too succinct, especially as we believe that the ability to do pathway searches is one of the main contributions of this work. To address this point, we have now updated the manuscript by including a better explanation of the pathway searches. Within the revised Appendix, we also include the complete list of 100 pathways tested together with the information on whether the intended target was recovered or not  (60% of the time). In addition, we are adding a one-page pdf that contains an example of a pathway search, with a complete visualization of the reaction tree leading to the target product.
>
> **Answers to the questions**
> * The RMechDB article (reference 25) introduces a data set with a corresponding database and web server for radical mechanistic reaction steps. It does not involve any reaction modeling or prediction.  However, as a service to the machine learning community, RMechDB comes with a standard train-test data split that can be used by anyone to compare different methods. We have added this information to Section 3 of the revised manuscript.
> * All the tables report the top N metrics (e.g. N=1, N=2, N=5, N=10) for the corresponding prediction in particular for single-step prediction. These are the standard metrics in the field (Schwaller, Philippe, et al 2019, Coley, Connor W., et al 2017, Irwin, Ross, et al. 2022, Tu et al 2023). As an intuitive example, the significance of the top N metric for single-step prediction (Table 4) is best understood in the context of the pathway (sequence of single steps) search problem. For instance, with a pathway search of length 3, a top 5 metric of 90% results in a 73%=(90%)^3 probability of recovering the target with a tree with a branching factor of 5 provided the pathways considered are congruent with the training set. Additional results are presented in Figure 3 where we assess the robustness of the predictors with respect to reaction type (left) and reactant size (right) using the top-5 metric, which are often important considerations in practical applications. For instance, atmospheric reaction predictions require a predictor to be robust with respect to the six radical reaction types described in the paper, as all six types occur frequently in the atmosphere. We have clarified all these points in the revised version.
> * Regarding the reviewer’s question in the third bullet above, the short answer is yes, with a caveat regarding the size of the training datasets. The RMechDB dataset consists of 5500 reactions only, which is in general too small for training a large text model from scratch. However, as stated in Section 4.4, we are able to use a pre-trained model (pre-trained on the US PTO dataset) and fine-tune it using the RMechDB dataset. This Molecular Transformer (Schwaller, Philippe, et al 2019)  approach is capable of predicting radical mechanistic reactions and provides intermediate byproducts. However, the existing text-based models do not predict orbital interactions (i.e. arrow codes). Furthermore, the pretraining is done on the reactions in the US PTO data set which are often non-balanced and report only the major product, resulting in biased predictions.
>
> **Comments on limitations**
> * We agree with the reviewer that the limitations of our work are not sufficiently addressed.  In the revised version of the manuscript, we have added a section discussing the limitations of our reaction predictor. In particular, we itemize the limitations in the following three categories:
> 1. Limited training data: RMechDB, as the first public database of radical mechanistic reactions, includes only 5500 mechanistic reactions.  Although these reactions are carefully curated, developing more advanced machine learning models such as LLMs typically requires datasets that are several orders of magnitude larger.
> 2. Limited range of radical chemistry: RMechDB, is focused on textbook reactions and atmospheric phenomena. This focus may affect the generalization capability of the proposed predictor in completely different areas that involve radical chemistry.
> 3. Limited search methods: Here, we used breadth-first search to search and expand the pathway tree of reactions. Although this approach guarantees the exploration of all the possibilities within the reaction tree, more intelligent search methods [Agostinelli, Forest, et al. 2021] may be capable of speeding up pathway searches.

---

> > ### Comment · Reviewer_G4db · 2023-08-14
> > **Thank for the additional details**
> >
> > The authors have clarified many of my major concerns and I have adjusted my score accordingly.

---

> > > ### Author Response · Authors · 2023-08-17
> > >
> > > We appreciate the reviewer's constructive comments and their willingness to adjust their score. We welcome further input to enhance our paper's quality in the time ahead.

---

### Official Review · Reviewer_PKyZ · 2023-07-07

**Soundness:** 3 good
**Presentation:** 2 fair
**Contribution:** 3 good
**Rating:** 6
**Confidence:** 4

**Summary:**

The authors provide a reaction predictor system that provides an accurate and interpretable prediction of radication reactions. Due to the lack of training data, there is a dearth of reaction predictors for radication reactions. The authors present 3 deep-learning-based approaches. The first approach is a two-step process that identifies possible reactive sites and then ranks the reactive site pairs. The second approach uses a contrastive learning approach to identify the most reactive site pairs. Finally, the authors also show a transformer-based approach to perform sequence-to-sequence translation from products to reactants. In the two-step, OrbChain approach, the authors present a GNN-based approach to identify reactive sites and a siamese network-based approach to rank the plausible reactive sites. Multiple reaction representations to perform plausibility ranking. The contrastive learning approach also uses a GNN and both a custom atom pair representation and a hypergraph representation are evaluated. Finally, a pre-trained MolGPT on USPTO dataset is used as well. The authors find that the graph-based methods outperform MolGPT and the contrastive learning methods yield the most accurate results.


**Strengths:**

- The authors compare multiple models to show the efficacy of different types of models such as GNNs and text-based Transformers for reaction prediction
- The authors also use multiple representations and model architectures for a very thorough evaluation of the proposed reaction


**Weaknesses:**

- It is not clear how or which of the three algorithms described is used in RMechRP.
- The presentation of the paper could be improved. There are 3 approaches described with multiple models and representations for some approaches. A short summary of the findings and comparisons or a visualization of the approaches could significantly improve the presentation


**Questions:**

- Line 100: How is the arrow-pushing mechanism A represented in OrbChain?
- In Table 2, what is the Atom Fingerprint method? Morgan fingerprint? ECFP?
- What is the loss function for the contrastive learning approach? (Might be in the appendix)
- Link 334: contrstive -> contrastive?


**Limitations:**

- Limited comparison as the authors don’t include a related works section so it is difficult to contextualize the scope of their current work to the field.
- Is there a reason the MolGPT model could not be trained on the new RMechDB dataset for evaluation rather than only fine-tuning?

---

> ### Author Rebuttal · Authors · 2023-08-10
>
> We appreciate the comments from the reviewer. We have addressed most of their comments in the revised version of our paper.
>
> **Comments on the weaknesses**
> * We agree that it is not obvious which model was used for RMechRP. The model used is the best combination of the two-step prediction method (as shown in Table 4), which provides the best compromise between speed, accuracy, and chemical interpretability. We have clarified this point in the revised version in Section 5.3.
> We agree with the reviewer that the number of models and representations might be confusing. To clarify this point, we have prepared a new figure that we have added to the revised manuscript (see also attached pdf).
>
> **Answers to the questions**
> * The arrow-pushing mechanism is represented using the numbers associated with the atoms and bonds involved in the arrow-pushing mechanism. It is best to illustrate this with an example.
> Reaction SMIRKS: CC(C)[O:10][N+:20]([O-])=O.[Ar]>>CC(C)[O:10].[O-][N+:20]=O.[Ar]
> Arrow codes: 10,20-10;10,20-20
> The arrow codes above are representing two arrows (separated by ‘;’). The first arrow starts at the bond between atom 10 and atom 20 and ends at atom 10. The second arrow starts at the bond between atom 10 and atom 20 and ends at atom 20. This information is already explained in the original RMechDB article (reference 25 in the manuscript) which we now cite also in line 100 to clarify this point.
> * Within Table 2, Atom Fingerprints refers to the method that is described in line 138. To avoid any confusion with other kinds of fingerprints, we have changed the name of this method to Atom Descriptor at line 138 and we have revised Table 2 accordingly.
> * The loss function of the contrastive learning approach is provided in line 48 of the appendix (Equation 1). For even greater clarity, we have also added this equation to Section 4.3.1 of the revised main manuscript.
> * In addition to the typo in line 334, we have fixed all the remaining typos in the main manuscript and the appendix in the revised version.
>
> **Comments on the limitations**
> * As the reviewers noticed, we merged the related work and introduction into one section where we reviewed the other predictors. As stated in the second paragraph of that section, most of the reaction predictors are operating at the level of multi-step transformations, not at the level of single mechanistic steps. To the best of our knowledge, the only other reaction predictor focused on predicting mechanistic steps is the one by Bradshaw et. al. 2018, which however is focused on non-radical reactions. So, our proposed reaction predictor is the only predictor with the scope of predicting radical reactions at the mechanistic level.
> * Other text-based models can be trained on the RMechDB data. However, it is essential to realize that RMechDB consists of a relatively small set of 5500 mechanistic reactions. Large language models (e.g., MolGPT) require a much larger number of training samples. While we did try to train an LLM from scratch, we found that it did not work well and thus adopted the more practical strategy of pretraining the LLM on the larger US PTO dataset, followed by fine-tuning it on the RMechDB dataset.

---

> > ### Author Response · Authors · 2023-08-19
> >
> > As the deadline is approaching, we are keen to ensure that our responses addressed all the concerns raised in your review.
> >
> > Based on your valuable feedback, we have made substantial revisions to improve the quality and clarity of our submission. Therefore, could you kindly consider updating your review or score to reflect these improvements?
> >
> > If there are still any unresolved concerns or areas that need further clarification, please let us know so that we can address them promptly.

---

> > > ### Comment · Reviewer_PKyZ · 2023-08-21
> > >
> > > Thank you for the clarification. I believe that I will stay with my current rating of 6.

---

> > > > ### Author Response · Authors · 2023-08-21
> > > >
> > > > We appreciate the reviewer's constructive comments. We believe addressing these comments has led to significant improvements in both quality and clarity of our submission.

---

### Official Review · Reviewer_URj3 · 2023-07-20

**Soundness:** 4 excellent
**Presentation:** 3 good
**Contribution:** 4 excellent
**Rating:** 7
**Confidence:** 5

**Summary:**

- a new model is described for prediction of radical chemical reactions
- the model is trained on a dedicated database of radical reactions for atmospheric chemistry, an important application
- several, reasonable baselines are evaluated

**Strengths:**

- reasonable, state of the art ML modelling (contrastive learning, attention GNNs, reasonable reaction representations inspired from molecular orbitals, building on previous work by Baldi's group)
- reasonable strong baselines (transformers)
- compelling results
- important application

**Weaknesses:**

- other baselines, like MEGAN https://pubs.acs.org/doi/abs/10.1021/acs.jcim.1c00537 or https://www.nature.com/articles/s42256-022-00526-z could be considered


### Related work
Several references in the introduction are not correct:

The Cao & Kipf MolGAN paper should be removed, because it does not deal with chemical reactions.
similarly, the Rogers et al ECFP does not deal with reaction prediction, and should be removed in the intro.

On the other hand, the Segler et al paper should be cited as an ML paper.

The ELECTRO paper by Bradshaw et al should be added. https://arxiv.org/abs/1805.10970

contrastive learning to distinguish between plausible and implausible reactions has already been used in https://www.nature.com/articles/nature25978 (called in-scope filter there), which should be referenced as well

**Questions:**

no questions, this is a solid, straightforward paper in my opinion

---

> ### Author Rebuttal · Authors · 2023-08-10
>
> We appreciate the fair comments from the reviewer and the fact that they acknowledged the importance of this work. We agree with all the suggested changes. We have revised our manuscript by implementing all the comments from this reviewer. Specifically, within the revised draft, we have removed the Cao & Kipf MolGAN and Rogers et al (ECFP) papers from the introduction. In addition, we have added the two suggested references (Bradshaw et al. 2018 and Siegler et al. 2018) to the introduction. Furthermore, we also cite Segler et al. again in Section 4.3 which is on contrastive learning.

---

> > ### Comment · Reviewer_URj3 · 2023-08-20
> >
> > Thank you!

---

### Official Review · Reviewer_n7xM · 2023-07-21

**Soundness:** 3 good
**Presentation:** 2 fair
**Contribution:** 3 good
**Rating:** 6
**Confidence:** 4

**Summary:**

Authors present two models that predicts radical chemistry reactions. The first model 'OrbChain' is comprised of two components 1) one GNN model for predicting pairs of reacting atoms/groups, 2) a model which ranks the plausibility of these pairs. The second model is a a fine-tuned Rxn-Hypergraph model, adapted for the task of predicting radical mechanism by using an atom classifier model.



**Strengths:**

+ Selects an interesting problem domain, specifically radical based chemistry.
+ Authors plan to open source the Radical Mechanistic Prediction model and release software for easier use.
+ Compares against relevant baselines, such as fingerprint representations, MolecularTransformer.

**Weaknesses:**

The presentation of results could be more clear. In particular:
- It would be helpful to have a clear statement of the key contributions provided by this work. The list of desiderata provided at the end of section 2 are important, but I believe that these properties have already been provided by previous models, especially references [20, 21] for radical reactions.
- If I understand correctly, OrbChain is the name of the two part model, but components of the model are still used for the second modeling approach using the fine tuned Rxn-Hypergraph model.
- The table formatting makes the results somewhat difficult to parse, it would be helpful to have more spacing between the caption and the table, and for the
- Table 3, where there is one column with 'AP \n Morgan2 \n TT' and it wasn't immediately obvious that these are different molecular descriptors.
-  For Figure 3, I believe the reaction type should be 'Homolysis' rather than homolyze.
- For Figure 3, It would be helpful to have a sense of the number of reactions in each class to better compare the relative performance by the model between reaction classes.
- There are several typos in the manuscript, e.g. a missing close parenthesis in lines 48-49 of page 2, 'weather' instead of 'whether' on line 188 on page 5, some tense mismatches. Please review for grammar errors.


In Section 2, authors state that 'None of the currrent reaction predictors can offer ... chemical interpretability, pathway interpretability, or balanced  atom mapping'. There are actually several models that provide interpretability for reaction mechanisms/reaction type. In addition to the works on radical mechanism prediction cited by the authors as references 20 and 21:

- In https://arxiv.org/pdf/1805.10970.pdf, Bradshaw et al. predict electron pair pushing mechansims with a generative model.
- The MolecularTransformer model has also been shown to provide atom mapping by visualizing attention weights (https://arxiv.org/pdf/2012.06051.pdf, Figure 2)./

For text based models such as Molecular Transformer, could you quantify the percentage of reaction predictions that suffer from a 'balance problem'? It isn't clear to me that this is a big issue with MolecularTransformer or other text based models.






**Questions:**

- Could you provide more information on the train test split used in the RMechDB? Are there any splits that are used to measure generalizability of predictions to out of domain reactions (e.g. by reaction type, atom types, structure similarity)

- Since interpretability is one of the benefits highlighted by this modeling approach, it would be interesting to see more examples of mechanism prediction by the proposed model in the main text.

- For the Pathway Search task, the results say that 60% of the reactants were found in the expanded reaction trees (of 10 step mechanisms). How well do current non-ML techniques perform on this task? How many of the proposed reactants are false positives; are false positive predictions of reactants detrimental to the problem prediction?

- What is the final model used in the RMechRP Software?

**Limitations:**

I do not identify any negative societal implications.

By my understanding, the presented model is intended to be limited for only radical based reactions, as it is trained on this domain of reactions, and not for other types of chemical reactions.

---

> ### Author Rebuttal · Authors · 2023-08-10
>
> We appreciate the comments from the reviewer. We have addressed most of their comments in the revised version of the manuscript.
>
> **Comments on the weaknesses**
> * We agree that references [20, 21] followed a similar approach toward reaction prediction. Nonetheless, our reaction predictor marks a substantial advancement across various facets, including training data, reaction modeling, and machine learning techniques. Notably, while [20, 21] present radical predictors built upon basic machine learning models trained on a limited dataset comprising only 96 radical reactions. These references cannot provide chemical and pathway interpretability.
> * It is important to note that OrbChain is not a prediction method. As explained in Section 4.1, OrbChain is the name of the method we developed to model a radical mechanistic reaction based on idealized molecular orbitals. Using OrbChain as a tool for modeling mechanistic reactions, we are able to develop the first two prediction methods called: two-step prediction and Contrastive learning.
> * Within the revised manuscript, we have fixed the spacing between the caption and the table to improve readability.
> * Within the revised manuscript, we have added information on AP, Morgan2, and TT used for the reactionFP method in Section 4.2.2.
> * About Figure 3, the reviewer is correct regarding the name of the reaction type. Within the revised manuscript, we have changed “Homolyze” to “Homolysis”.
> * In Figure 3, the number of reactions in each class is shown by the blue bar which is labeled “All Reactions”.
> * In addition to all the typos mentioned by the reviewer, we have fixed all the remaining typos in both the main manuscript and the Appendix.
> * In Section 2, where we delve into the advantages of our proposed reaction prediction method, we introduced the concepts of "chemical interpretability" and "pathway interpretability." Based on our definitions, existing reaction prediction models fall short of delivering these two benefits. This deficiency arises because these models are either not trained with orbital level information (for chemical interpretability) or lack training on mechanistic reaction data (for pathway interpretability). Although the current reaction prediction models can offer interpretability within the context of machine learning (e.g., attention weights), it's important to distinguish this from the chemical and pathway interpretability as per our definitions. The third advantage of our reaction prediction model lies in its ability to maintain balance. The concern of balance preservation is seldom addressed in models trained on USPTO data, given the inherent imbalance of the data source. Therefore, the qualitative assessment of the reviewer that the balance effect “is not a big issue” lacks substantiating evidence. Conversely, when applying a reaction predictor to specific real-world scenarios like drug degradation and mass spectrometry, preserving the balance is an essential consideration. Since our reaction predictor is capable of predicting balanced reactions, it has the potential to be applied to a range of problems where the use of current reaction prediction models is not practical.
>
>
> **Answers to the questions**
> * The RMechDB article (reference 25) released an online platform of radical mechanistic reactions with splits into train and test sets. Quoting from the RMechDB paper, “the reaction data are carefully split into train and test data where the test set replicates the distribution of the reaction type and reactive orbitals of the training data”. Also, RMechDB data (both train and test sets) are extracted from the radical chemistry textbooks and research articles on atmospheric chemistry. Therefore, there is no test set to explicitly measure the generalization capability of the model for out-of-domain reactions.
> * We agree that the description of the pathway search experiment is too succinct, especially as we believe that the ability to do pathway searches is one of the main contributions of this work. To address this point, we have now updated the manuscript by including a better explanation of the pathway searches. Within the revised Appendix, we also include the complete list of 100 pathways tested together with the information on whether the intended target was recovered or not  (60% of the time). In addition, we are adding a one-page pdf that contains an example of a pathway search, with a complete visualization of the reaction tree leading to the target product.
> * The final model used in the RMechRP software is the best combination of the two-step prediction method to provide the orbital and pathway interpretability. Within the revised draft, we have added this information to Section 5.3.

---

> > ### Comment · Reviewer_n7xM · 2023-08-11
> >
> > Thank you authors for their explanations and efforts to make the work more accessible. The addition of Figure 1 in the attachment is very helpful for laying out the problem and the proposed solutions. I think that the authors could do more to expand either Figure 1 or the caption to Figure 1 to better highlight the contributions by this work.
> >
> > Based on my understanding of your responses and rereading the paper, my understanding of the main contributions of this work are:
> >
> > - Benchmarking several methods on the radical specific dataset RMechDB (ref. 25). The transformer model, RxnHypergraph are previously published models, but authors here introduce new features to make use of the RxnHypergraph model. Of the proposed three methods, only the two-step prediction method provides both chemical interpretability (i.e. an assignment of molecular orbitals), but all methods contribute pathway interpretability.
> >
> > - OrbChain is a new GNN model presented in this work that can 1) Classify reaction types, 2) Predict reaction outcomes using the reaction types. The method here is somewhat similar to the reaction prediction algorithms published ref. 20 (Learning to Predict Reactions). I do not follow the authors' point in the rebuttal about how the model in this reference does not provide chemical interpretability -- If I understand this work correctly, I believe specific one-step mechanisms are predicted by the model in the form of identifying electron 'sources' and electron 'sinks' and solving a matching problem. If I understand correctly, the main difference with OrbChain is that it is a GNN model, and covers a much wider scope of reactions because of the use of RMechDB as training; is this correct?
> >
> > With the changes made by the authors to help clarify the contributions of this work, and the changes proposed by the authors to improve readability/typos, I have edited scores given in my original review.
> >
> > Question specifically about Figure 1 in the attachment:
> > - What is meant by OrbChain generating 'Labels'? Does this mean reaction classification labels?

---

> > > ### Author Response · Authors · 2023-08-15
> > >
> > > We thank the reviewer for their constructive comments. We also appreciate their willingness to adjust their score. Using their second set of comments we have further revised our manuscript.
> > >
> > > **Regarding Figure 1 in the attachment**
> > >
> > > * We agree with the reviewer that the caption is perhaps too concise. As a result, in the revised version, we have expanded the caption to read:
> > > “This is a schematic depiction of the prediction problem, the processing tool (OrbChain), and the three approaches. The three approaches are: Two Step Prediction, Contrastive Learning, and Text-Based. The first two approaches use OrbChain to find the reactive orbitals for training and to form the products during inference.”
> > >
> > > **Regarding the first question on OrbChain**
> > >
> > > * We wish to clarify that OrbChain is not really a GNN, but rather a reaction processing tool that is used by the GNN models (see Equation 1 describing OrbChain). This processing step assigns labels  to molecular orbitals and their atoms before training the GNNs. This also answers the second question about OrbChain raised by the reviewer: OrbChain provides labels at the level of orbitals and atoms, not at the level of reactions.
> > > The text-based models do not need OrbChain because they operate directly on the text representations, not the orbitals. This reviewer is correct in stating that the use of the RMechDB dataset significantly expands the scope of the reactions.

---

### Author Rebuttal · Authors · 2023-08-10

This one-page pdf file includes three figures that are generated to improve the clarity of the paper and also to provide a better response to the reviewers' comments.

---

### Decision · Program_Chairs · 2023-09-21

**Decision:**

Accept (poster)

**Comment:**

Radical chemistry is an exciting application area for reaction prediction models due to its focus on elementary steps, understanding of which is an achilles heel of currently used deep networks. As agreed by most reviewers, the paper has solid modeling and establishes useful benchmark results for the task. The main weakness of the paper is its lack of clarity. In particular, it is challenging to understand that OrbChain model is not actually a neural network. Its also challenging to draw meaningful conclusions from the multi-step results section. All in all, I am happy to recommend acceptance. I would like to ask the Authors to try to further improve clarity in the camera ready version.